# Challenges of conducting an international observational study to assess immunogenicity of multiple COVID-19 vaccines

Ratna Sardana[1‡]*, Placide Mbala Kingebeni[2‡], Wiwit Agung Snc[3], Abdoul H. Beavogui[4], Jean-Luc Biampata[2], Djeneba Dabitao[5], Paola del Carmen Guerra de Blas[6], Dehkontee Gayedyu-Dennis[7], Mory C. Haidara[4], Ganbolor Jargalsaikhan[8], Garmai Nyuangar[7], Asep Purnama[9], Guillermo Ruiz Palacios[10], Seydou Samake[5], Moctar Tounkara[5], Shera Weyers[11], Delgersaikhan Zulkhuu[8], Sally Hunsberger[1], Renee Ridzon[1]

1 Division of Clinical Research, National Institute of Allergy and Infectious Diseases, National Institutes of Health, Bethesda, Maryland, United States of America, 2 Institut National de Recherche Biomédicale, Kinshasa, Democratic Republic of the Congo, 3 Rumah Sakit Umum Daerah Dr. H. Moch. Ansari Saleh Hospital, Banjarmasin, Indonesia, 4 Centre National de Formation et de Recherche en Santé Rurale de Maferinyah, Maferinyah, Guinea, 5 University Clinical Research Center, University of Sciences, Techniques and Technologies of Bamako, Bamako, Mali, 6 The Mexican Emerging Infectious Diseases Clinical Research Network, Mexico City, Mexico, 7 Partnership for Research on Vaccines and Infectious Diseases in Liberia, Monrovia, Liberia, 8 The Liver Center, Onom Foundation, Ulan Bator, Mongolia, 9 TC Hillers Hospital, Maumere, Indonesia, 10 Departamento de Infectología, Instituto Nacional de Ciencias Médicas y Nutrición Salvador Zubirán, Mexico City, Mexico, 11 Clinical Monitoring Research Program Directorate, Frederick National Laboratory, Frederick, Maryland, United States of America

‡ RS and PMK are co-first authors on this work.
* rsardana@niaid.nih.gov

**Data Availability Statement:** All relevant data are within the paper.

## Abstract

The International Study on COVID-19 Vaccines to Assess Immunogenicity, Reactogenicity, and Efficacy is an observational study to assess the immunogenicity of COVID-19 vaccines used in Democratic Republic of Congo, Guinea, Indonesia, Liberia, Mali, Mexico, and Mongolia. The study, which has enrolled 5,401 adults, is prospectively following participants for approximately two years. This study is important as it has enrolled participants from resource-limited settings that have largely been excluded from COVID-19 research studies during the pandemic. There are significant challenges to mounting a study during an international health emergency, especially in resource-limited settings. Here we focus on challenges and hurdles encountered during the planning and implementation of the study with regard to study logistics, national vaccine policies, pandemic-induced and supply chain constraints, and cultural beliefs. We also highlight the successful mitigation of these challenges through the team's proactive thinking, collaborative approach, and innovative solutions. This study serves as an example of how established programs in resource-limited settings can be leveraged to contribute to biomedical research during a pandemic response. Lessons learned from this study can be applied to other studies mounted to respond rapidly during a global health crisis and will contribute to capacity for stronger pandemic preparedness in the future when there is a crucial need for urgent response and data collection.

**Funding:** This project has been funded in whole or in part with federal funds from the National Cancer Institute, National Institutes of Health, under Contract No. 75N91019D00024. The content of this publication does not necessarily reflect the views or policies of the Department of Health and Human Services, nor does mention of trade names, commercial products, or organizations imply endorsement by the U.S. Government. This research was supported by the Intramural Research Program of the NIAID/NIH; no grant money was received from any outside source. The funders collaborated with the participating countries on study design, data collection and analysis, decision to publish, and preparation of the manuscript.

**Competing interests:** The authors have declared that no competing interests exist.

## Background/aims

The emergence of SARS-CoV-2 led to the COVID-19 pandemic, with a wide spectrum of manifestations ranging from asymptomatic infection to severe illness with acute respiratory distress syndrome and death [1]. Vaccines have been developed that are safe and effective at preventing serious illness, hospitalization, and death from COVID-19. The differences in immunogenicity of different vaccines can be influenced by the vaccine type, vaccine delivery platform, host factors (e.g., age, sex, nutritional status, immune status, comorbidities) and infrastructure issues such as capacity to sustain cold chain, administration, scheduling, and time since vaccination [2–6].

The International Study on COVID-19 Vaccines to Assess Immunogenicity, Reactogenicity, and Efficacy (InVITE) was launched in August of 2021 and is still ongoing (https://clinicaltrials.gov/ct2/show/NCT05096091). InVITE is a study to assess the immunogenicity of COVID-19 vaccines used per national guidelines through the national immunization programs of seven low- and middle-income countries, with the aim of informing the global response to the pandemic regarding findings potentially related to or impacting the choice of vaccines, implementation of vaccination programs, policy for the use of booster doses of vaccines and documented SARS-CoV-2 infection rate after vaccination [7].

Although InVITE is one of several observational studies conducted during a public health emergency, there are unique aspects to this study. A total of 5,401 adults have been enrolled, and the study is prospectively following participants for approximately two years, a period longer than many other vaccine immunogenicity studies.

There is a need for diversity of settings for biomedical research with special attention to inclusion of resource-limited settings. The seven low- and middle-income countries on three continents participating in this trial—Democratic Republic of Congo (DRC), Guinea, Indonesia, Liberia, Mali, Mexico, and Mongolia—are culturally, ethnically, and geographically diverse. According to www.clinical trials.gov, there are multiple COVID-19 vaccine clinical trials registered for Mexico and Indonesia. However, for DRC, Guinea, Mali and Mongolia, there is only one registered in each country and none in Liberia, so there is little or no data on immune responses to available initial or booster COVID-19 vaccines in these countries.

There are significant challenges to mounting a study during an international health emergency. Here we focus on some challenges and hurdles that were successfully mitigated by the team's proactive thinking and collaborative approach.

The study aim is to evaluate immune responses to vaccines given in each country, across countries for the same vaccines, and in specified subgroups defined by age, body mass index, pregnancy, comorbidities, HIV infection, other co-infections including malaria, or prior infection with SARS-CoV-2. To minimize lab-to-lab variation in assay results, samples from all sites are shipped to a central laboratory in the U.S., and testing is performed using standardized assays [8]. In addition to the scheduled study visits, study participants are asked to return for a symptomatic visit throughout the study if they develop symptoms consistent with SARS-CoV-2 infection, at which time a blood sample and an upper airway swab are obtained. For participants with SARS-CoV-2 infection, viral genomic sequencing is performed to characterize strains and understand the impact of waning immunity and viral variants on vaccine effectiveness [7].

## Challenges experienced

We categorized challenges into study logistics, national vaccine policies, pandemic-induced and supply-chain constraints, and cultural beliefs.

## Study logistics

A strength of InVITE is that it is taking place in seven different countries that all have experience conducting clinical studies. However, the clinical researchers involved with the study have varying levels of scientific expertise and experience. Meanwhile, accounting for country-specific regulatory policies, informed consent requirements, standard operating procedures, and other needs presented a challenge to the initiation and execution of the study.

## National vaccine policies

While the aim of InVITE is to examine the immunogenicity of COVID-19 vaccines, vaccination itself is not provided by the study. Instead, each country's national immunization program provided the vaccines. Type of vaccine and naïve and booster regimens differed by country. This reliance created challenges for recruitment and follow up. As enrollment was required within 24 hours of vaccine administration, the study team needed to collaborate with the immunization program staff to enroll participants during this tight time frame. Vaccine distribution strategies within the participating countries varied from being very slow due to, for instance, poor attendance at a vaccination site, to the teams being inundated with participants if outreach programs were implementing effective vaccination campaigns to meet vaccination targets. In addition, it was difficult to schedule follow-up visits and blood draws for two-dose regimens since the timing of follow-up visits was based on completion of a vaccination regimen. The study needed to rely on anticipated timing of the second vaccine and on participants' report of receiving the second vaccine, the timing of which was dependent on factors outside the study team's control, such as vaccine availability, distribution schedules and adherence with a follow-up vaccine visit.

During the study, emerging evidence of vaccine-induced adverse events and suspension of certain vaccines in several European countries resulted in changes to some countries' policies regarding the number and timing of booster vaccines [9]. In Democratic Republic of Congo, use of the AstraZeneca vaccine was suspended for a period after reports of vaccine-associated blood clots were issued, leading to reluctance of the population to get vaccines and a change in the vaccination distribution strategy. To best capture data that reflected implementation practices in each country, the study design required flexibility to accommodate and incorporate the different and changing practices in each setting, such as restricting vaccine during pregnancy and in those with certain medical conditions such as hypertension.

## Pandemic-induced and supply chain constraints

Travel restrictions for the U.S. study team prevented conducting on-site feasibility assessments, in-person training and site monitoring visits. This lack of face-to-face interaction along with multiple time zone differences made it uniquely challenging to address site questions or concerns. The pandemic also impacted the study supply chains that were required to create and maintain the sample biorepositories and to ship study samples to the central laboratory. The laboratory teams across the sites had to manage laboratory supply shortages due to the global disruption of supply chains. Many study supplies that were not available in-country had to be shipped from the U.S. without guarantee of timely delivery due to limited supply, flight shortages, customs requirements, and import costs. Shipment of archived samples from the

country sites to the central laboratory in the U.S. was hampered by limited options for international couriers specialized in transporting biological materials and a scarcity of supplies needed for shipping, including dry ice and International Air Transport Association-recommended shipping materials.

## Cultural beliefs

Rumors and misconceptions about COVID-19 vaccines, such as the effect of vaccines on fertility, had a major impact on study enrollment. In addition, potential study participants' concerns related to the volume of the blood drawn, future use of samples collected, use of vaccines during pregnancy and vaccine effectiveness were not easy to dispel. As the pandemic evolved, there was also concern that a perception of decreased virus threat would lead to poor attendance at scheduled study visits and symptomatic visits.

## Innovative solutions

The InVITE team generated processes and procedures to overcome the aforementioned challenges and managed a successful and speedy study implementation, enrolling the first participant just five months after the study concept was developed (Fig 1). Below are some of the innovative solutions used to overcome the obstacles we faced.

## Study logistics

Strong in-country leadership was essential to uniformly implement the study across all sites while managing country-specific needs and regulations. Each country established a local study team that was supported with remotely conducted training and regular teleconferences with the central U.S.-based team. By allowing each country to develop and submit its own site-specific appendix along with the main protocol for country regulatory review, InVITE provided flexibility to accommodate country-specific regulatory policies. This flexibility allowed changes in the vaccine regimens being used, including those using two different vaccines in a two-dose initial vaccine regimen. In some countries, such as Mexico, regulatory pathways and requirements were altered in response to the COVID-19 pandemic, and the scientific teams and regulatory entities had to collaborate to obtain regulatory approvals. Flexibility was also required to accommodate country-specific projected vaccine delivery and distribution timelines, informed consent requirements and standard operating procedures. For example, in Mongolia, government policy initially declared that a single booster dose was to be administered; however, this policy was updated to include receipt of a second booster dose, and the flexibility in the protocol to accommodate this allowed the enrollment of those receiving 1st and 2nd booster doses of vaccine. A second example is that recruitment needed to be tailored to the setting(s) in each country where vaccines were delivered because the local vaccination programs administered vaccines in a variety of locations, including hospitals, clinics, and mobile vans. As a result of this flexibility, only a single protocol amendment was needed during the study. In an effort to replicate in-person training that could not occur due to pandemic-induced travel restrictions, the study team developed an "InVITE Study Simulation Video" and other pictorial and written educational materials to enhance understanding of the protocol. In response to evidence of waning immunity, the protocol was amended to include additional booster doses of vaccine and extend the study follow-up with specimen collection at two additional study visits. Study data were entered into a central database that was reviewed with generation of queries if questions arose or if case report form (CRF) completions were overdue.

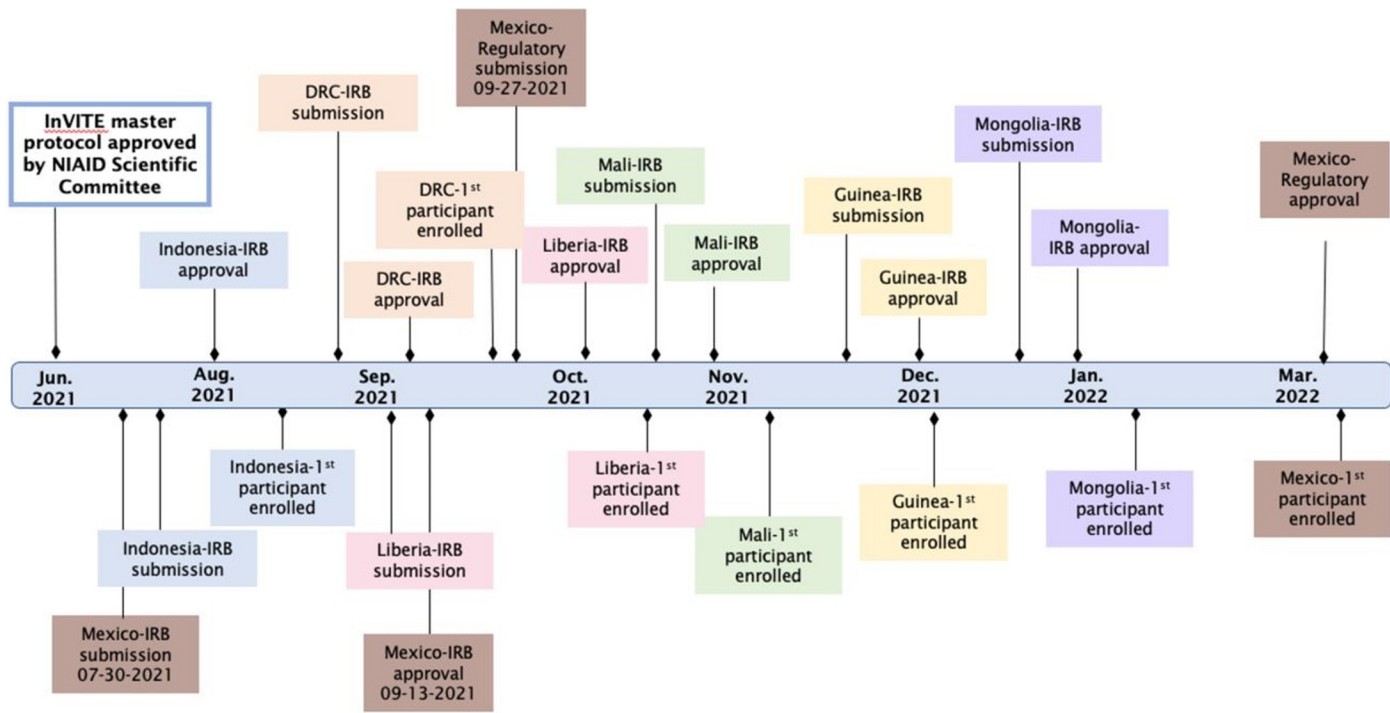

**Fig 1. Timeline of InVITE study implementation.** IRB-Institutional Review Board; NIAID-National Institute of Allergy and Infectious Diseases.

## National vaccine policies

The study teams closely monitored their national vaccine distribution timelines and communicated with each country's national immunization program to maintain study enrollment, understand upcoming policy changes, and stay abreast of data management. This required study teams to know when to take advantage of vaccine campaigns to boost enrollment or to pause or slow enrollment so that country teams could catch up on CRF completion.

To accurately capture information (such as the introduction of additional vaccine booster doses or testing algorithms to diagnose SARS-CoV-2 infections based on national policies), CRFs were updated as needed and site-specific forms were created. Community health workers, who are members of the community, were trained to provide informal counseling, perform outreach efforts, and serve as study advocates. They were engaged as part of the study team to make phone calls and home visits to remind participants of follow-up visits. Some tailored solutions were implemented by partnering with government health authorities. For example, one country team provided participants with the results of SARS-CoV-2 serology tests performed locally as incentive to participate in the study. Another country requested access to the vaccination schedule from their local health authorities so that reminders could be sent to participants to complete their vaccine regimens. There were situations that made it difficult for participants to attend follow-up visits such as extended religious holidays, including Ramadan and the Lunar New Year, or travel outside of a study area due to military duty or employment. To avoid missed visits, the window periods for each visit are wide (1–2 months). Despite reminders, some participants were unable to return for visits within the study windows. To maximize specimen and data collection, an SOP was developed to allow for out-of-window visits emphasizing the importance of collecting visit samples outside of the original 1–2-month window while remaining within the protocol requirements.

## Pandemic-induced and supply chain constraints

A team conducted monthly inventory checks to ensure sufficient supplies at all site labs and to schedule shipments, keeping in mind that additional time was required due to in-country customs clearance delays and COVID-19 regulations. Frequent communication among the laboratories, logistics team, study team and shipping couriers helped resolved issues quickly. To

**Table 1.**

| Issue | Challenge | Solution(s) |
|---|---|---|
| Study Logistics | Study did not provide vaccines | Close coordination with immunization programs to track vaccine delivery. Frequent contact with participants to remind of need to complete vaccination with (re) scheduling of study visits based on completion of vaccination. |
| | Variation in country-specific needs and requirements | Each country submitted a site-specific appendix to the main protocol for regulatory approval to address and fulfill their site-specific regulatory policies and procedures. |
| | Keeping participants motivated to continue study visits | Contact with participants through phone calls or visits by community health workers to remind them of scheduled visits. |
| | Missed or late study visits | Allowance for study visits outside of the allotted study visit window, classified as an "out-of-window" visit, and development of Standard Operating Procedures for handling these visits within the study requirements. |
| | Shipment of samples from countries that had not worked with international couriers transporting biological specimens | Frequent ongoing communication between study teams and couriers. Test shipment to confirm that the viability of samples was not compromised during shipping. A temperature monitor and GPS tracker were included in each shipment to track any temperature deviation and prevent loss of shipment during transit. |
| National Vaccine Policies | Changing vaccine policies with addition of booster doses | Close communication with local health authorities to keep abreast of upcoming changes in regimens and amending the protocol as needed. |
| | Delayed vaccine schedules due to supply constraints | Close communication between country team and data management team to establish the accurate timing and data collection forms for follow-up visits based on changes and delays in vaccine schedule. |
| | Mixed vaccine regimens due to interrupted supply of one type of vaccine | Flexibility in protocol to allow mixed vaccine regimens for study participants. |
| Supply Chains and Pandemic-Induced Constraints | Interrupted supply chains | Assign a dedicated team to track supplies at each site, identify vendors, schedule shipments based on level of enrollment at country sites and time required for custom clearances. |
| | Restricted travel leading to remote training | Develop study simulation videos and conduct remote training for study staff on protocol implementation. |
| Cultural Beliefs | Vaccine hesitancy | Community engagement sessions to educate and inform potential participants. |
| | Concerns about amount of blood collected | Clear explanation of the quantity of blood collected using common measures. |

avoid delays in getting pre-assembled kits from international suppliers, local teams were remotely trained in kit assembly, and an SOP with pictures was developed. Due to a shortage of Whatman cards needed for dried blood spots, study staff temporarily stored whole blood samples until replacement cards arrived. Meanwhile, some countries had not worked with international couriers transporting biological materials to the U.S. before; in these instances, a test shipment confirmed that sample viability was not compromised during the handling, packaging, and shipping and that all procedures were understood and applied according to the shipping SOP. A temperature monitor and GPS tracker were included in each shipment to track any temperature deviation and prevent loss during transit.

## Cultural beliefs

Community health workers helped educate participants and dispel vaccine-related fears, rumors, and doubts. These individuals were key to successfully achieving the study enrollment goals. To support recruitment efforts, the study team developed illustrative charts and brochures that explained the study purpose, duration, procedures, rationale for the volume of blood drawn, benefits, and risks.

Because participants were encouraged to come in for scheduled study visits as well as symptomatic visits, dedicated resources and study staff were needed for making regular follow-up calls to participants. To mitigate the stigma and implications of receiving a positive diagnosis (including missing work), intensive community engagement was conducted to educate and inform potential participants about the study and the need to come for a visit if they developed symptoms of COVID-19. In some countries, community health workers were responsible for tracking participants and encouraging adherence to study visits. It is likely that this tracking contributed to the very high compliance with follow-up visits thus far. Table 1 shows challenges the InVITE team faced and the solutions that were deployed.

## Discussion

The InVITE study data from low- and middle-income countries will be an important addition to information on immunogenicity of COVID-19 vaccines in settings that have not been well represented in pandemic-related clinical research. Additionally, the InVITE study has served as a learning opportunity for clinical researchers from the seven participating countries who have varying levels of scientific experience and capabilities. This experience with study planning, regulatory submission, training, and study initiation and implementation during an ongoing pandemic has contributed to research capacity and pandemic preparedness at each of the study sites.

Implementation of the study was not without challenges that needed attention and innovation to overcome. Close coordination and communication among members of the entire team led to timely initiation of the InVITE study and implementation without any major setbacks or protocol violations. Involvement of community health workers contributed to high compliance with scheduled and symptomatic study visits.

For large trials like InVITE to be implemented in diverse geographic, social, cultural and research settings, it is important to establish core leads (data management, laboratory, regulatory compliance, research coordination, community engagement, and study principal investigators) from each country to work with the subject matter experts so that perspectives from all collaborating teams are appreciated, challenges are recognized early, and solutions are promptly implemented during both study planning and implementation. Because it was important to initiate the study rapidly and conduct it in the ever-changing landscape of the pandemic, the time taken to generate appropriate training tools and conduct test shipments of

samples was well spent. Bringing all the key in-country and central team members together regularly on investigator calls was important for information sharing and remains an essential activity, especially for large multi-country studies. Although it is a general practice to train study staff how to engage and educate study participants, we learned that the addition of community health workers was essential for participant enrollment and retention.

Despite obstacles faced by the InVITE team, a collaborative effort has enabled the study team to address challenges that arose during the study. This study serves as an example of how established programs in resource-limited settings can be leveraged to contribute to biomedical research during a pandemic response. Lessons learned from this study can be applied to other studies mounted to respond rapidly during a global health crisis and will contribute to capacity for stronger pandemic preparedness in the future when there is a crucial need for urgent response and data collection.

## Acknowledgments

The study team would like to thank Esther Akpa, Naranbaatar Dashdorj, Olivier Tshiani Mbaya, Abelardo Montenegro-Liendo, Mary Smolskis, and Moctar Tounkara for their help and support.

## Author Contributions

**Conceptualization:** Placide Mbala Kingebeni, Abdoul H. Beavogui, Jean-Luc Biampata, Djeneba Dabitao, Paola del Carmen Guerra de Blas, Dehkontee Gayedyu-Dennis, Guillermo Ruiz Palacios, Seydou Samake, Moctar Tounkara, Shera Weyers, Sally Hunsberger, Renee Ridzon.

**Data curation:** Sally Hunsberger, Renee Ridzon.

**Formal analysis:** Ratna Sardana, Abdoul H. Beavogui, Jean-Luc Biampata, Djeneba Dabitao, Paola del Carmen Guerra de Blas, Ganbolor Jargalsaikhan, Garmai Nyuangar, Asep Purnama, Shera Weyers, Delgersaikhan Zulkhuu, Sally Hunsberger, Renee Ridzon.

**Funding acquisition:** Ganbolor Jargalsaikhan, Sally Hunsberger.

**Investigation:** Ratna Sardana, Placide Mbala Kingebeni, Wiwit Agung Snc, Abdoul H. Beavogui, Jean-Luc Biampata, Djeneba Dabitao, Paola del Carmen Guerra de Blas, Dehkontee Gayedyu-Dennis, Mory C. Haidara, Garmai Nyuangar, Asep Purnama, Shera Weyers, Delgersaikhan Zulkhuu, Sally Hunsberger, Renee Ridzon.

**Methodology:** Ratna Sardana, Djeneba Dabitao, Paola del Carmen Guerra de Blas, Shera Weyers, Sally Hunsberger, Renee Ridzon.

**Project administration:** Placide Mbala Kingebeni, Wiwit Agung Snc, Dehkontee Gayedyu-Dennis, Mory C. Haidara, Ganbolor Jargalsaikhan, Garmai Nyuangar, Asep Purnama, Guillermo Ruiz Palacios, Seydou Samake, Moctar Tounkara, Shera Weyers, Delgersaikhan Zulkhuu, Sally Hunsberger, Renee Ridzon.

**Software:** Shera Weyers.

**Supervision:** Placide Mbala Kingebeni, Abdoul H. Beavogui, Jean-Luc Biampata, Dehkontee Gayedyu-Dennis, Mory C. Haidara, Ganbolor Jargalsaikhan, Guillermo Ruiz Palacios, Seydou Samake, Moctar Tounkara, Shera Weyers, Sally Hunsberger, Renee Ridzon.

**Validation:** Guillermo Ruiz Palacios, Renee Ridzon.

**Visualization:** Ratna Sardana, Guillermo Ruiz Palacios, Renee Ridzon.

**Writing – original draft:** Ratna Sardana, Placide Mbala Kingebeni, Jean-Luc Biampata, Djeneba Dabitao, Paola del Carmen Guerra de Blas, Guillermo Ruiz Palacios, Shera Weyers, Renee Ridzon.

**Writing – review & editing:** Ratna Sardana, Placide Mbala Kingebeni, Wiwit Agung Snc, Abdoul H. Beavogui, Jean-Luc Biampata, Djeneba Dabitao, Paola del Carmen Guerra de Blas, Dehkontee Gayedyu-Dennis, Mory C. Haidara, Ganbolor Jargalsaikhan, Garmai Nyuangar, Asep Purnama, Guillermo Ruiz Palacios, Seydou Samake, Moctar Tounkara, Shera Weyers, Delgersaikhan Zulkhuu, Sally Hunsberger, Renee Ridzon.

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
