## [Decision Letter · Decision Letter 0]

21 Mar 2023

PGPH-D-23-00316

Challenges of conducting an international observational study to assess immunogenicity of multiple COVID-19 vaccines

Dear Ratna,

Thank you for submitting your manuscript to PLOS Global Public Health. After careful consideration, we feel that it has merit but does not fully meet PLOS Global Public Health’s publication criteria as it currently stands. Therefore, we invite you to submit a revised version of the manuscript that addresses the points raised during the review process.

We look forward to receiving your revised manuscript.

Kind regards,

Collins Otieno Asweto, PhD

Academic Editor

Journal Requirements:

1. Please amend your detailed online Financial Disclosure statement. This is published with the article. It must therefore be completed in full sentences and contain the exact wording you wish to be published.

a) State the initials, alongside each funding source, of each author to receive each grant.

2. In the Funding Information you indicated that no funding was received. Please revise the Funding Information field to reflect funding received.

Please ensure that the funders and grant numbers match between the Financial Disclosure field and the Funding Information tab in your submission form. Note that the funders must be provided in the same order in both places as well.

3. Please provide separate main figure files in .tif or .eps format only and ensure that all files are under our size limit of 10MB.

Reviewers' comments:

Reviewer's Responses to Questions

**Comments to the Author**

1. Does this manuscript meet PLOS Global Public Health’s publication criteria? Is the manuscript technically sound, and do the data support the conclusions? The manuscript must describe methodologically and ethically rigorous research with conclusions that are appropriately drawn based on the data presented.

Reviewer #1: Partly

Reviewer #2: Yes

2. Has the statistical analysis been performed appropriately and rigorously?

Reviewer #1: Yes

Reviewer #2: N/A

3. Have the authors made all data underlying the findings in their manuscript fully available (please refer to the Data Availability Statement at the start of the manuscript PDF file)?

Reviewer #1: Yes

Reviewer #2: Yes

4. Is the manuscript presented in an intelligible fashion and written in standard English?

Reviewer #1: Yes

Reviewer #2: Yes

5. Review Comments to the Author

Reviewer #1: • What are the main claims of the paper and how significant are they for the discipline?

This paper focused on challenges and hurdles encountered during the planning and implementation of the study with regard to study logistics, national vaccine policies, pandemic-induced and supply chain constraints, and cultural beliefs. It also highlights the successful mitigation of these challenges through the team’s proactive thinking, collaborative approach, and innovative solutions. In the context of a pandemic, an international study across 3 continents on the challenges of COVID-19 vaccination most especially in low resource setting is very significant

• Are the claims properly placed in the context of the previous literature? Have the authors treated the literature fairly?

The claims of this paper are succinctly presented and well addressed.

• Do the data and analyses fully support the claims? If not, what other evidence is required?

The only data presented (graphically) is the submission of enrolees by country and date. The number and demographics of enrolees by country was not presented.

• If the paper is considered unsuitable for publication in its present form, does the study itself show sufficient potential that the authors should be encouraged to resubmit a revised version?

This paper does not address the entirety of its topic. The aim of this paper as stated was to evaluate/assess immunogenicity, reactogenicity and efficiency of COVID-19 vaccine among the selected countries through an observational study, however these key issues were not addressed in the write up. Hence the paper does not speak to the Topic.

There is need to bring in the immunogenicity, reactogenicity and the efficiency of COVID-19 vaccines as it relates to the enrolees in this study, as this will add more to the body of knowledge regarding the complexities surrounding COVID-19 vaccination more especially in low- and middle-income countries.

To retain this selected topic, there is need to present finding around the 3 key issues mentioned above or the Topic can be modified to focus on Challenges of COVID-19 vaccinations and not immunogenicity.

• Is the manuscript well organized and written clearly enough to be accessible to non-specialists - YES

Reviewer #2: Sharing of lessons learnt is encouraged to support similar studies in the future, making future studies time and cost-efficient.

Comments

• Line 56,97 and others: The full stop is before the brackets. Check throughout the document for this issue.

• Line 203: Introduce who community health workers are. This cadre is not available in every country or is known using a different terminology therefore, readers may not be familiar.

• Line 248: It just says SOP- Standard Operating Procedures. The other columns for challenges and solutions are not filled.

• Line 256 – 257: Speaks about capacity gained. Describe the capacity gained. well described.

• Table 1: In the study logistics column, it says that the challenge is “Study did not provide vaccines”. The study is an observational study meaning there is no intervention therefore, it is not expected that there would be the provision of vaccines by the project.

• Table 1: Study logistics column “Variation in country-specific needs and requirements”. This is not clear. Does it mean in terms of ethics/research permits or that implementation had to be tailored to country-specific needs? Link to the wording in the paragraph on study logistics in lines 109-114.

• General comments:

o It is useful for readers of these types of manuscripts to learn detail on how sites gained capacity during the conduct of this study. It provides insight to the usefulness of these studies and shows the preparedness of these sites to attract future studies.

o The narrative is well written but it is missing specific examples that would be useful to the reader. Example lines 182-184 “Flexibility was also required to accommodate country-specific projected vaccine delivery and distribution timelines, informed consent requirements and standard operating procedures”. What is meant by flexibility? Perhaps also giving an example of a specific country on how it was worded. Lines 205-207 are a good example of a specific description of how a specific challenge was accommodated.

• Apart from cultural beliefs, were any challenges faced with observation of religious events or seasonality such as rains? How were these challenges overcome?

• When was the study conducted?

• During the period of the project were any of the countries under lockdown, curfews or restricted movement? If yes, how were these challenges overcome for follow-up?

6. PLOS authors have the option to publish the peer review history of their article (what does this mean?). If published, this will include your full peer review and any attached files.

**Do you want your identity to be public for this peer review?** For information about this choice, including consent withdrawal, please see our Privacy Policy.

Reviewer #1: **Yes: **Dr. AHMED MAMUDA BELLO

Reviewer #2: No

---

## [Decision Letter · Decision Letter 1]

25 May 2023

Challenges of conducting an international observational study to assess immunogenicity of multiple COVID-19 vaccines

PGPH-D-23-00316R1

Dear Sardana,

We are pleased to inform you that your manuscript 'Challenges of conducting an international observational study to assess immunogenicity of multiple COVID-19 vaccines' has been provisionally accepted for publication in PLOS Global Public Health.

Best regards,

Collins Otieno Asweto, PhD

Academic Editor

Reviewer Comments (if any, and for reference):

Reviewer's Responses to Questions

**Comments to the Author**

1. If the authors have adequately addressed your comments raised in a previous round of review and you feel that this manuscript is now acceptable for publication, you may indicate that here to bypass the “Comments to the Author” section, enter your conflict of interest statement in the “Confidential to Editor” section, and submit your "Accept" recommendation.

Reviewer #1: All comments have been addressed

2. Does this manuscript meet PLOS Global Public Health’s publication criteria? Is the manuscript technically sound, and do the data support the conclusions? The manuscript must describe methodologically and ethically rigorous research with conclusions that are appropriately drawn based on the data presented.

Reviewer #1: Yes

3. Has the statistical analysis been performed appropriately and rigorously?

Reviewer #1: Yes

4. Have the authors made all data underlying the findings in their manuscript fully available (please refer to the Data Availability Statement at the start of the manuscript PDF file)?

Reviewer #1: Yes

5. Is the manuscript presented in an intelligible fashion and written in standard English?

Reviewer #1: Yes

6. Review Comments to the Author

Reviewer #1: All comments have been addressed.

7. PLOS authors have the option to publish the peer review history of their article (what does this mean?). If published, this will include your full peer review and any attached files.

**Do you want your identity to be public for this peer review?** For information about this choice, including consent withdrawal, please see our Privacy Policy.

Reviewer #1: No
